# Biological Activities of Rhamnan Sulfate Extract from the Green Algae *Monostroma nitidum* (Hitoegusa)

**DOI:** 10.3390/md18040228

**Published:** 2020-04-24

**Authors:** Koji Suzuki, Masahiro Terasawa

**Affiliations:** 1Faculty of Pharmaceutical Sciences, Suzuka University of Medical Science, Minamitamagaki-cho, Suzuka, Mie 513-8670, Japan; 2Konan Chemical Manufacturing Co., LTD., Kitagomizuka, Kusu-cho, Yokkaichi, Mie 510-0103, Japan; terasawa@konanchemical.co.jp

**Keywords:** *Monostroma nitidum*, rhamnan sulfate, antithrombotic effect, antivirus, vascular endothelium dysfunction

## Abstract

*Monostroma nitidum* is a green single-cell layered algae that grows on the southwest coast of Japan. It is often used for salad ingredients, boiled tsukudani, soups, etc., due to its health benefits. *M. nitidum* is composed of many cell aggregates, and the various substances that fill the intercellular space are dietary fibers, vitamins, and minerals. Rhamnan sulfate (RS), a sulfated polysaccharide, is main the component of the fiber extracted from *M. nitidum*. Recently, some biological properties of RS have been demonstrated by in vitro and in vivo studies that probably protect human subjects from viruses and ameliorate vascular dysfunction caused by metabolic disorders, especially lifestyle-related diseases. In this review, we focus on the antithrombotic effects of RS and introduce its antiviral and other biological activities.

## 1. Introduction

Seaweeds have traditionally been consumed by people inhabiting coastal areas in East Asia. Seaweed is rich in vitamins and minerals and is known to have many favorable effects on human health. There are three groups of seaweed, namely green algae, red algae, and brown algae. Green algae contain large amounts of chlorophyll, or green pigments such as those found in terrestrial plants. In contrast, red algae contain, in addition to chlorophyll, plenty of red pigments called phycoerythrin and blue pigments called phycocyanin, giving them a red or purple color [1,2]. Brown algae contain, in addition to chlorophyll, red–brown pigments called fucoxanthin and take on a yellowish brown or blackish brown color. The green alga *Monostroma nitidum* is often confused with a species of sea lettuces (*Ulva* Linnaeus). However, *M. nitidum* and sea lettuce belong to a different order in biological classification. Many species in the group of sea lettuces have a double layer of cells, whereas *M. nitidum* have a single layer of cell assemblies (Figure 1a,b) [3]. The Japanese name of *M. nitidum* Hitoegusa is derived from the fact that the leaf-like tissue is composed of a single layer of cells. *M. nitidum thallus* consists of many cell aggregates (Figure 1c), separated with mucilage rich in dietary fiber, vitamins, minerals, lipids, and peptides [3].

*M. nitidum* grows wild in shallow waters along the Pacific coast of Japan (Kii Peninsula, Kyushu, and Nansei Islands), Korean Peninsula, and Southern China. In Japan, seaweed farming is also being practiced in the southwest coastal areas of Mie, Kumamoto, Kagoshima, Nagasaki, and Okinawa prefectures. It is particularly popular in Matoya Bay, Ago Bay, Ise Bay, and Gokasho Bay in Mie Prefecture, accounting for about 70% of the total production in Japan. *M. nitidum* is a health-promoting food that contains dietary fibers consisting of polysaccharides, minerals, and proteins, and it has been used extensively for nori boiled in soy sauce, miso soups, and salads. 

## 2. Biochemical Characteristics of Rhamnan Sulfate

Cell walls of *M. nitidum* contain a large amount of rhamnan sulfate (RS), a sulfated polysaccharide [4,5]. Although this macromolecule is poorly soluble in water, it can be solubilized and decomposed into smaller molecules by treating *M. nitidum* with hot water. The obtained extract is purified by anion-exchange resin column chromatography; sulfated substances bind to the column and are subsequently eluted from the column by increasing the concentration of sodium chloride. This method produces a highly purified RS [5]. Figure 2a shows hot water extract from the intercellular substance of *M. nitidum*, and Figure 2b shows highly purified RS from *M. nitidum*.

Purified RS is a macromolecular polysaccharide with a molecular weight of tens of thousands to millions. It consists of a polysaccharide with long linear chain structures of α-1,3-linked l-rhamnose connected with α-1,2-linked branched chains, to which several sulfate groups bind (Figure 3) [6,7]; sulfate groups account for about 25% of *M. nitidum*-derived RS [8]. 

## 3. Biological Activities of RS

RS has been reported to have biological activities such as antiviral [6,9,10,11,12,13], anticoagulant [14,15,16,17,18,19,20,21], and antitumor [22] activities. It also induces hyaluronidase inhibition [23], exhibits anti-hypercholesterolemic effect [24,25] suppresses blood glucose elevation [26], and helps with weight control [25]; thus, *M. nitidum*-derived RS is considered to be involved in many activities in host defense of humans. Table 1 is a summary of the biological activities of RS reported to date. 

In what follows, we describe some of the biological activities of *M. nitidum*-derived RS that we have studied and reported, especially its antiviral, anti-obesity, anti-hypercholesterolemic, anti-glycemia, antithrombotic, and anti-inflammation effects. 

### 3.1. Antiviral Effect of RS

Medical researchers have hitherto developed various antiviral medications, including acyclovir, an inhibitor of DNA synthesis used as a medication for herpes simplex virus type 1 and type 2, which cause strong inflammatory responses in the skin, mouth, lips (herpes labialis), eyes, and sex organs, and another medication called oseltamivir, a neuraminidase inhibitor of the influenza virus. However, shortly after their commercialization, these antiviral medications were met with the emergence of drug-resistant viruses, posing a major challenge. It is known that, because these antiviral agents specifically inhibit enzymes of the viruses, they are more likely to induce mutations in the viral DNA or RNA. To overcome this problem, researchers sought to identify and analyze novel substances with antiviral properties that are extracted from naturally derived substances. RS was the outcome of one of those studies [9].

In vitro analysis of the antiviral effect of RS revealed that RS showed suppressing effects on human immunodeficiency virus (HIV), human cytomegalovirus (HCMV), and herpes simplex virus types 1 and 2 (HSV-I and -II) [6,9,10]. Recently, Terasawa et al. showed that RS has an antiviral effect on enveloped viruses, such as measles virus, mump virus, influenza A virus, and human corona virus, but does not affect non-enveloped viruses, such as adenovirus, poliovirus, coxsackie virus, and rhinovirus. Evaluation of the in vitro anti-influenza A virus activity of RS showed that it inhibited both virus adsorption and entry steps to Madin-Darby canine kidney (MDCK) cells [13]. 

Wang, et al. have shown that intramuscular injection of RS into enterovirus-infected mice significantly improves their survival rate, and RS directly binds to viruses and thereby prevents viral absorption to tissue cells [12], suggesting that it can exert antiviral effects in vivo. Terasawa et al. also showed that oral administration of RS in influenza A virus–infected mice suppressed viral proliferation and weight loss and stimulated neutralizing antibody production in vivo. RS inhibited virus adsorption to the host cells. RS probably binds to the influenza virus hemagglutinin (HA), which is a glycoprotein on the viral envelope and important for host cell binding. Both antiviral and immune stimulating activities of RS may contribute to the suppression of viral proliferation in vivo [13].

### 3.2. Anti-Obesity and Anti-Hypercholesterolemic Effects of RS

Seaweed contains large amounts of dietary fibers and is reported to lower serum cholesterol levels and neutral fats through carrageenan [27]. Accordingly, to clarify whether the extracted RS, a soluble fiber, is effective for treating obesity and obesity-related diseases, including hepatic steatosis, diabetes, and hyperlipidemia, the zebrafish was used to assess and examine its effects. The zebrafish is a model organism that has been used in recent years to identify and assess drug candidate agents for vertebrates. When zebrafish were rendered obese after being fed a high-fat diet, they doubled their weight compared to normal feeding in four weeks. When switching to a diet containing RS, the weight gain of these obese zebrafish was significantly attenuated [25]. RS also significantly (*p* < 0.05) mitigated the elevation of serum LDL cholesterol induced by a high-fat diet and prevented fat accumulation in the liver. The gene expression analysis showed that RS mainly inhibits lipogenesis. These results suggest that RS has anti-obesity effects. We conducted a clinical trial on 16 adult males with high cholesterol levels [24]. Their serum cholesterol levels were determined after the study subjects took RS every day for six weeks. As a result, there was no change in serum HDL cholesterol level, but LDL cholesterol level in the RS-treated group was significantly (*p* < 0.05) reduced compared with that of the group not taking RS. These results suggest that RS keeps the LDL/HDL cholesterol ratio low and prevents lifestyle-related diseases such as obesity and hyperlipidemia. 

### 3.3. Anti-Glycemia Effect of RS

Kamimura et al. [26] reported the effect of edible green algae *M. nitidum* powder and RS on glycemic responses after oral administration of various saccharinity agents to rats. When rats were administered 2 g of glucose, sucrose, maltose, or soluble starch/kg body weight (b.w.) of rat with or without *M. nitidum* powder (200 mg/kg b.w. of rat) or RS (approximately 50 mg/kg b.w. of rat), both *M. nitidum* powder and RS significantly inhibited the increment of plasma glucose level, compared to the controlled group (*p* < 0.01). Furthermore, they evaluated the effects of *M. nitidum* powder and RS on the postprandial increase in blood glucose level in healthy human subjects. The results showed the significant deduction of the blood glucose level at 30 min after the intake of *M. nitidum* powder (500 mg) or RS (approximately 60 mg) compared to the controlled ones (*p* < 0.05). These data suggest that the *M. nitidum* powder and RS are useful for diabetes mellitus.

In general, dietary fiber has the effect of suppressing an increase in blood glucose level [28,29], and the mechanisms thereof include (i) delay in the transfer of carbohydrate from the stomach to the duodenum, and (ii) inhibition of carbohydrate digestion and absorption in small intestine. Recently, Terasawa found that RS also has activity to inhibit α-amylase, an enzyme that digests starch to sugar (personal data). The activity may also contribute suppressing an increase in blood glucose level.

### 3.4. Antithrombotic Effects of RS 

Risk factors for the development of thrombosis include aging, various environmental factors, clinical conditions, and stress. Thrombosis is a clinical condition that occurs when a pathological thrombus develops in the blood vessel. Platelet aggregation, blood clotting, and vascular endothelial inflammation are alleged to be three major factors closely related to thrombus formation in blood vessels (Virchow’s triad for thrombosis) (Figure 4) [30,31]. Rudolf K. Virchow in 1856 proposed that there are three essential factors contributing to the formation of arterial and venous thrombosis. They are pathological changes in (1) blood flow, such as abnormal increase and decrease of shear stress; (2) blood components, such as abnormal activation of platelets, blood coagulation, and fibrinolysis factors; and (3) vessel wall, such as abnormal vascular endothelial inflammation. Even today, these three factors are important in understanding the formation of arterial and venous thrombosis, and in developing medicines to treat and prevent thrombotic disorders [32,33].

#### 3.4.1. Blood Coagulation and Platelet Aggregation under Vascular Endothelial Cell Inflammation 

It has been recognized that the most important factors contributing to the formation of pathological thrombi in the blood vessels are pathological changes, such as inflammation of the vascular endothelium [34,35]. Factors such as aging, obesity, hypertension, diabetes, various types of stress, and smoking induce chronic damage to the endothelium. In general, normal vascular endothelial cells express various molecules that inhibit thrombus formation. Examples include thrombomodulin (TM), heparan sulfate proteoglycan, tissue factor pathway inhibitor (TFPI), and protein S, which inhibit blood coagulation; nitric oxide (NO), prostacyclin (PGI_2_), and ecto-ATP/ADPase, which inhibit platelet activation; tissue plasminogen activator (t-PA), which enhances fibrinolysis; and anti-inflammatory cytokines, IL-4, IL-10, NO, etc., as listed in Table 2, are expressed [35]. 

In contrast, damaged endothelial cells have reduced almost all antithrombotic molecules and express many molecules that promote thrombus formation. Examples include tissue factor (TF), which triggers blood coagulation reaction; von Willebrand factor (VWF), which triggers platelet aggregation; plasminogen activator inhibitor-1 (PAI-1), which attenuates fibrinolysis; and tumor necrosis factor-α (TNF-α), which induces an inflammatory response (Table 3) [34,35,36]. 

These lead to the activation of platelets and leukocytes (neutrophils, monocytes, lymphocytes, etc.), stimulation of adhesion molecules, adhesion of leukocytes to the endothelium, and proliferation of vascular muscle cells. These changes lead to the development and progression of arteriosclerosis, unstable or ruptured plaques, the generation of a thrombin-induced coagulation thrombus, and ultimately to various types of thrombosis (Figure 5).

Endothelial injury induced by various factors promotes the expression of many molecules that may trigger hypercoagulation, attenuation of fibrinolysis, and increased inflammatory response, giving rise to arteriosclerosis, unstable or ruptured plaques, the generation of a thrombin-induced coagulation thrombus, and ultimately various types of thrombosis. 

In infectious diseases, various types of pathogen-derived molecules (pathogen-associated molecular patterns: PAMPs for short) are derived from extrinsic pathogenic microbes, which may lead to the production of inflammatory cytokines or molecules associated with the innate immune system [37,38]. For example, bacterial-membrane-derived lipopolysaccharides (LPS) cause acute damage to endothelial cells and body tissue cells [39]. Different from PAMPs, a group of cell-derived molecules (damage-associated molecular patterns: DAMPs for short) that are released in association with cell death or injury may lead to the production of inflammatory cytokines and molecules associated with the innate immune system [37,38]. For example, DNA and histone that are leaked out from injured or disrupted cells significantly induce inflammation in the endothelial cells, leading to the production of a large quantity of TF, VWF, and TNF-α [40]. This, in turn, accelerates blood coagulation and platelet aggregation, leading to the formation of microthrombi in the capillary of various organs and the induction of disseminated intravascular coagulation (DIC) [41], which may result in multiple-organ failure (MOF) [42] or severe bleeding. Moreover, hemolytic uremic syndrome (HUS) caused by *Escherichia coli* O-157 infection [43], sinusoidal obstructive syndrome (SOS)/veno-occlusive disease (VOD) [44,45], and transplantation-associated thrombotic microangiopathy (TA-TMA) [46] are all disorders or conditions that result from endothelial damage and are therefore symptoms of endothelial damage. To prevent the onset of these diseases, it is quite important to prevent the inflammation of the endothelium. 

#### 3.4.2. Platelet Aggregation Inhibitory Effect of RS

Formation of thrombi due to enhanced platelet aggregation is known to cause artery thrombosis [47,48]. To identify the effects that RS has on platelet aggregation, the effects of RS on human platelet aggregation in whole blood induced by collagen, thrombin, or an antibiotic ristocetin (one of the activators of VWF) were examined. As a result, as shown in Figure 6a–c, RS strongly inhibited collagen-, thrombin-, and ristocetin/VWF-induced platelet aggregation in a dose-dependent manner. RS has been shown to inhibit platelet aggregation in whole blood, and therefore is expected to be useful as a food for preventing arterial thrombosis. 

#### 3.4.3. Blood Coagulation Inhibitory Effect of RS

The formation of a pathological coagulation thrombus due to activation of the blood coagulation system in blood vessels is thought to cause mainly venous thrombosis. To examine the effect of RS on blood coagulation, human blood plasma was used to examine the effect of RS on the prothrombin time (PT), which measures extrinsic coagulation ability, and on the activated partial thromboplastin time (APTT), which measures intrinsic coagulation ability. Heparin was used to compare the difference with RS. Heparin is one form of sulfated polysaccharides, and it has been used as an anticoagulant from old times [49]. As shown in Figure 7a,b, RS prolonged both the PT and APTT in a concentration-dependent manner. This result shows that RS plays an inhibitory role in the extrinsic, as well as in the intrinsic, coagulation system [14]. However, the anticoagulation effect of RS was comparable to that of heparin at RS concentrations of approximately 10-fold the concentration of the latter. Thus, it can be inferred that the coagulation inhibitory effect of RS is about one-tenth that of heparin. It has been reported that intraperitoneally administered RS suppressed the PT and APTT in in vivo experiments using rats [16,17]. Therefore, RS is thought to be effective in preventing the development of venous thrombosis.

#### 3.4.4. Inhibition Mechanism of Blood Coagulation by RS and Heparin

The anticoagulant heparin exists in animal mast cells, lung, intestine, spleen, and liver. The sugar chain components of heparin are structurally heterogeneous, in which glucuronic acid binds to uronic acid or N-acetylglucosamine binds to iduronic acid, and these are bound by a large amount of sulfate groups [50]. It has been revealed that the underlying mechanism of heparin’s anticoagulant activity is not that heparin directly inhibits the coagulation factor, but that it first binds to antithrombin (AT) in the blood, to alter (activate) the three-dimensional structure of active site of AT, and this activated form of AT in turn binds to thrombin or factor Xa, to inhibit coagulation activity [51,52].

In clinical practice, unfractionated (polymeric) heparin (molecular weight 5000–20,000; extracted mainly from porcine intestinal mucosa), low-molecular-weight heparin (molecular weight 4000–6000; prepared by processing unfractionated heparin, using a sugar-chain degradation enzyme), and fondaparinux (molecular weight 1728; an ultra-low-molecular-weight heparin consisting of five sugars obtained from further chemosynthesis) have been used. Unfractionated (polymeric) heparin not only promotes the inhibition of factor Xa via AT, but also greatly enhances thrombin inhibition by binding directly to both AT and thrombin. In contrast, low-molecular-weight heparin and fondaparinux cannot bind to thrombin. Thus, they only promote the inhibition of factor Xa via AT. 

To elucidate the coagulation inhibitory mechanism of RS, its inhibitory effects on thrombin and factor Xa in the presence or absence of purified AT were analyzed. As shown in Figure 8a, the thrombin inhibitory effect was not observed in RS itself. However, similar to heparin, it inhibited thrombin activity in the presence of AT [14]. While heparin completely inhibited the activity of factor Xa in the presence of AT, RS inhibited only around 30% of factor Xa activity; the inhibitory effect of RS was thus weak compared to heparin (Figure 8b). The data show that RS strongly inhibits thrombin and weakly inhibits factor Xa in an AT-dependent manner. They also suggest that the anticoagulant effect of RS is driven by a molecular mechanism similar to that of unfractionated (polymeric) heparin. 

The reason why RS has less of an anticoagulant effect than heparin is probably as follows. The minimal structure of the AT binding site in heparin has been shown to be GlcNAc-GlcA-GlcN-IdoA-GlcN [53]. However, RS does not have a structure like the AT binding site of heparin. Therefore, the binding strength between RS and AT could be about 1/10 that of heparin. 

### 3.5. Anti-Inflammation Effect of RS on Endothelial Cells

It is important to prevent the endothelial inflammation to regulate the onset of thrombotic disorders. To identify the differences between RS and heparin in their effects on inflammation of endothelial cells, we examined whether RS regulates the expression of TF, a trigger of blood coagulation and inflammation, and the platelet aggregating factor VWF in bacteriogenic LPS-induced inflammation of endothelial cells. As shown in Figure 9a, adding LPS induced a marked elevation in TF expression in the endothelial cells, while the TF expression was significantly (*p* < 0.01) suppressed in the presence of RS in an RS-concentration-dependent manner [14]. Moreover, RS itself significantly suppressed (*p* < 0.01) the TF expression in endothelial cells not stimulated with LPS in a dose-dependent manner. Similarly, as shown in Figure 9b, LPS significantly induced the expression of VWF in the endothelial cells, and the VWF expression was significantly (*p* < 0.01) suppressed in the presence of RS in a concentration-dependent manner [14]. Moreover, RS significantly suppressed (*p* < 0.01) the VWF expression in the endothelial cells not stimulated with LPS in a dose-dependent manner. The effect of heparin was analyzed under the same conditions, and the results showed that, although heparin did not affect the TF expression in the endothelial cells stimulated with LPS, it significantly suppressed (*p* < 0.01) the TF expression in the endothelial cells not stimulated with LPS (Figure 9c). Unlike RS, heparin enhanced VWF expression in the endothelial cells both in the presence and absence of LPS (Figure 9d). Likewise, RS suppressed significantly the elevation of TF and VWF expression in TNF-α- or thrombin-induced inflammation of endothelial cells, whereas heparin did not show such a suppressing effect (data not shown here) [14]. Overall, unlike heparin, RS was shown to strongly attenuate inflammatory injury in cultured vascular endothelial cells. RS is suggested to be a novel anticoagulant and anti-inflammatory substance for vascular endothelial cells. 

## 4. Closing Remarks

*M*. *nitidum* RS has a wide range of health-promotion effects in lifestyle-related disorders—it prevents obesity and the elevation of LDL cholesterol and suppresses the elevation of blood glucose levels [24,25,26]. It was also shown to improve various vascular dysfunctions caused by lifestyle-related diseases. In addition to the health-promoting effects of lifestyle-related disorders, RS may be important in combating viral epidemics. Figure 10 shows the possibility that oral ingestion of RS prevents viral infection, thrombotic disease, hypercholesterolemia, hyperglycemia, and so on.

RS is present mainly in green algae *Ulva pertusa*. It has been reported that laver (Nori) in brackish waters contains a small amount of RS and *M. nitidum* contains large amounts of RS [54]. It is also found in the microalga spirulina [55]. However, differences in the structure and biological activity of RS between these species remain unclear. 

For biological activities observed in RS to be effective in humans, this macromolecule must be transported into the blood after its oral intake. As a polymer with a molecular weight of more than 30,000, RS transportation into the blood via oral ingestion has traditionally been considered difficult. Previous studies on fucoidan, a polymeric sulfate polysaccharide similar to RS, may be helpful for elucidating the in vivo absorption of RS. Fucoidan is a dietary fiber that is abundant in viscous matter of brown algae, such as kelp, seaweed, and mozuku (*Nemacystus decipiens*). It is mainly a sulfated polysaccharide composed of tens to hundreds of thousands of L-fucose, and its average molecular weight is about 200,000. In a clinical study of fucoidan, about 50 ng/mL of fucoidan was detected in the blood six hours after the single oral administration of the agent [56], demonstrating that fucoidan, despite its small quantity, was absorbed from the intestine and transported to the blood. Although fucoidan and RS are composed of different sugars, their chemical structures are similar, so it is assumed that RS enters the blood in a similar manner. Terasawa et al. showed that FITC-labeled RS colocalized with Peyer’s patch M cells 30 min after oral administration of the RS, indicating that RS could bind to the M cells and become incorporated into Peyer’s patches [13]. Microfold cells (M cells) are a part of the intestinal epithelium that is involved in the selective uptake of antigens, such as polymers and bacteria, from the gut lumen into Peyer’s patches, an intestinal lymphoid tissue [57,58,59]. Thus, it is quite possible that orally ingested RS is transported from the gut lumen into the blood via M cells in Peyer’s patches, where it carries out various biological activities.

With an increasing elderly population, patients with lifestyle-related diseases are estimated to increase in the near future, and how this increase could be prevented poses a major challenge. As is often said, “humans age as blood vessels age”; however, it is considered that RS is an attractive food for preventing lifestyle-related diseases and prolonging health and life expectancy.

## Figures and Tables

**Figure 1 marinedrugs-18-00228-f001:**
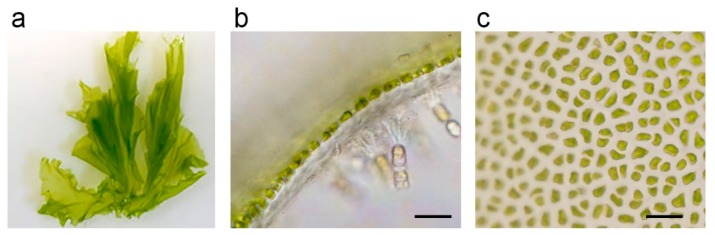
(**a**) *Monostroma nitidum*; (**b**) longitudinal section of the tissue, which is composed of a single layer of cells; (**c**) tissue consists of many cell aggregates; bar indicates 20 μm.

**Figure 2 marinedrugs-18-00228-f002:**
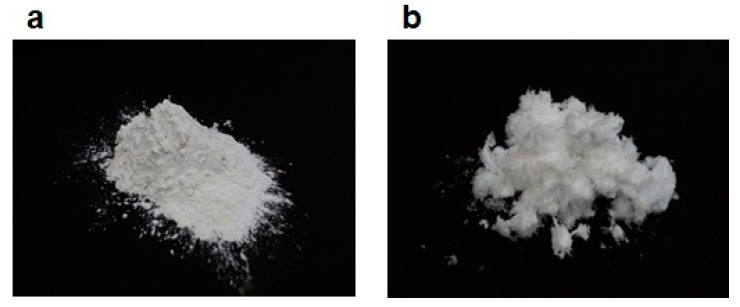
(**a**) Hot-water extract from intercellular substance of *M. nitidum*; (**b**) highly purified rhamnan sulfate from *M. nitidum*.

**Figure 3 marinedrugs-18-00228-f003:**
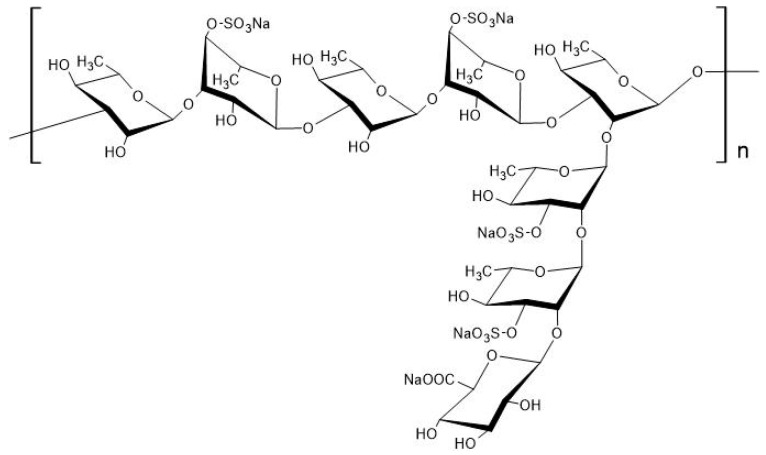
Chemical structure of rhamnan sulfate (RS) extracted from *Monostroma nitidum*. RS is a polysaccharide composed of linear chains of l-rhamnose (α-1,3 linkages) with branched chains (α-1,2 linkages), to which several sulfate groups are bound; reproduced from Tako et al. [7]. Reproduced with permission from SinencePG, 2020.

**Figure 4 marinedrugs-18-00228-f004:**
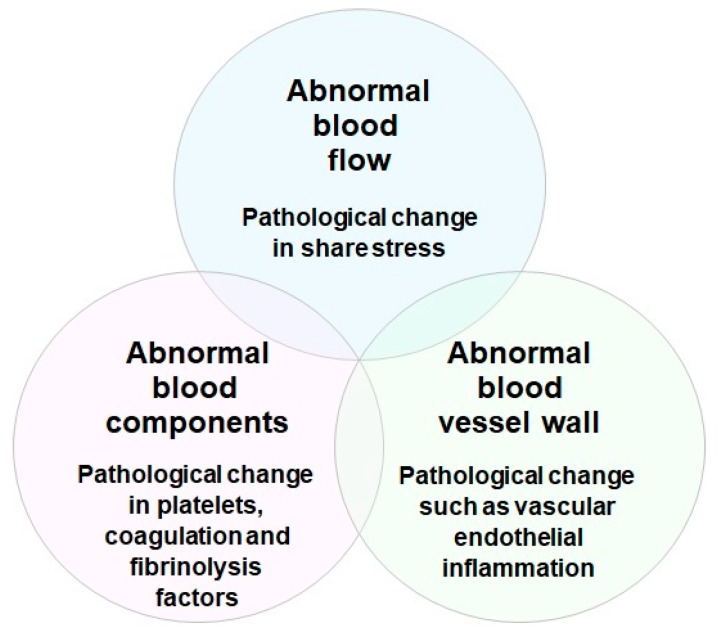
Virchow’s triad for thrombosis. In 1856, Rudolf K. Virchow proposed three essential factors contributing to the formation of arterial and venous thrombosis. They are pathological changes in blood flow (shear stress), blood components (platelets, coagulation, and fibrinolysis factors), and blood vessel wall (endothelial cell inflammation).

**Figure 5 marinedrugs-18-00228-f005:**
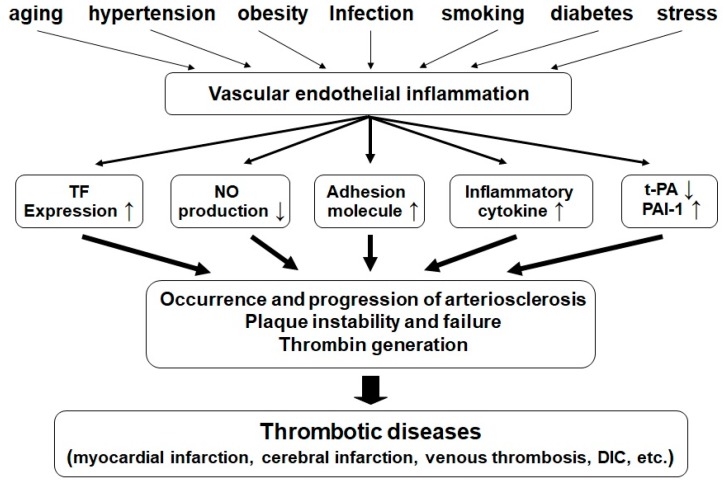
Schematic diagram of the mechanism of thrombotic disorders caused by vascular endothelial inflammation associated with lifestyle-related diseases and various clinical symptoms.

**Figure 6 marinedrugs-18-00228-f006:**
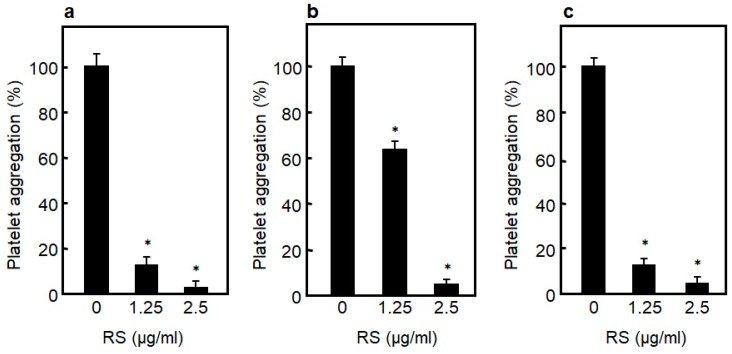
Effect of rhamnan sulfate (RS) on human platelet aggregation induced by (**a**) collagen, (**b**) thrombin, and (**c**) ristocetin. Concentrations of collagen, thrombin, and ristocetin to induce platelet aggregation are 0.3 mg/mL, 1.25 U/mL, and 1 mg/mL, respectively. Significant difference in values between the sample without RS and the sample with RS are shown as * *p* < 0.01. The data are cited from [14]. Reproduced with permission from Springer Nature, 2019.

**Figure 7 marinedrugs-18-00228-f007:**
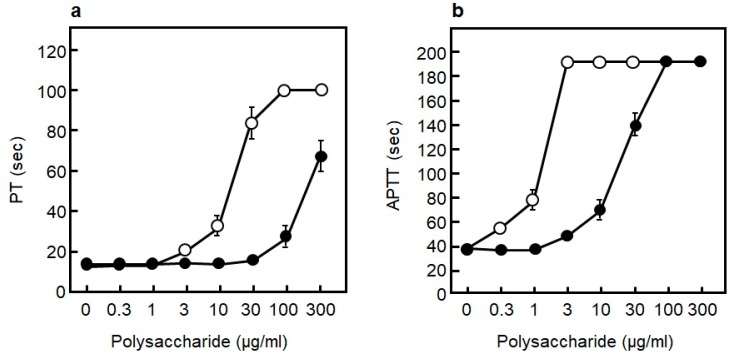
Effect of different polysaccharide on blood coagulation. (**a**) Effect of rhamnan sulfate (RS) (●) and heparin (**○**) on extrinsic coagulation measured by prothrombin time (PT). (**b**) Effect of RS (●) and heparin (**○**) on intrinsic coagulation measured by activated partial thromboplastin time (APTT). The data are cited from [14]. Reproduced with permission from Springer Nature, 2019.

**Figure 8 marinedrugs-18-00228-f008:**
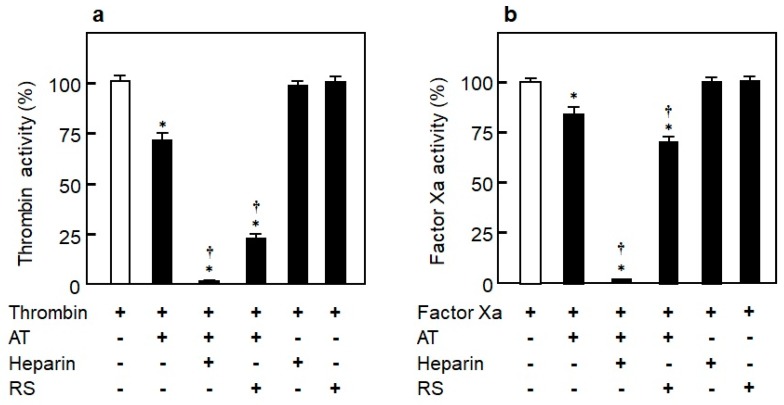
Mechanisms of rhamnan sulfate (RS) and heparin on inhibition of blood coagulation in the presence or absence of antithrombin (AT). (**a**) Effect of RS (2 μg/mL) and heparin (2 μg/mL) on inhibition of thrombin (2.5 nM) in the presence or absence of AT (5 nM). (**b**) Effect of RS (2 μg/mL) and heparin (2 μg/mL) on inhibition of factor Xa (12.5 nM) in the presence or absence of AT (25 nM). Significant difference in values between the sample with thrombin alone or factor Xa alone and the sample of thrombin or factor Xa with AT, heparin, or RS is shown as * *p* < 0.01. Significant difference in values between the sample including thrombin + AT or factor Xa + AT and the sample including thrombin + AT or factor Xa + AT with heparin or RS is shown as ^†^
*p* < 0.01. The data are cited from [14]. Reproduced with permission from Springer Nature, 2019.

**Figure 9 marinedrugs-18-00228-f009:**
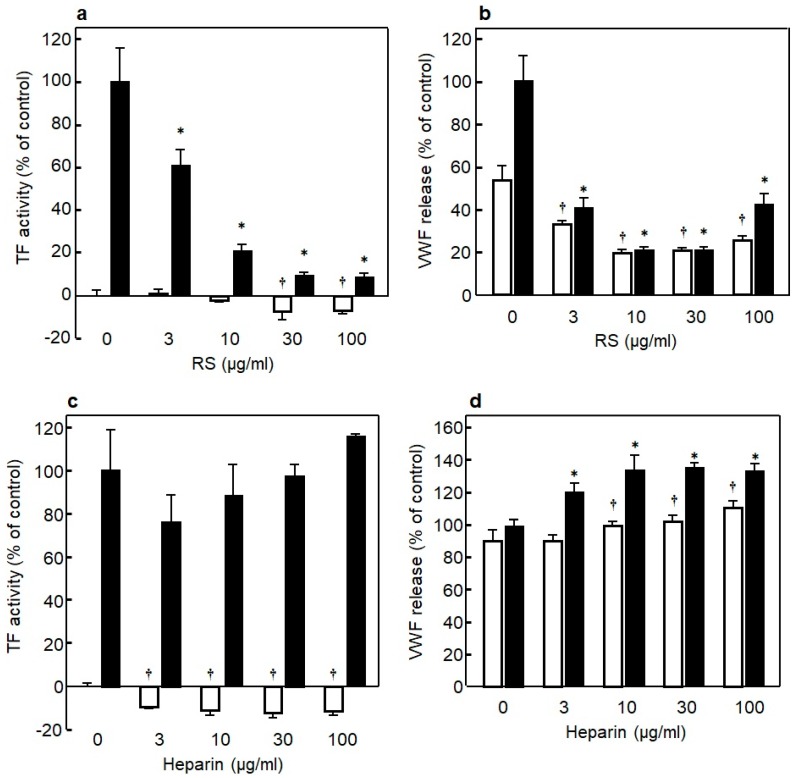
(**a**) Effect of various concentrations of rhamnan sulfate (RS) on tissue factor (TF) expression in HUVECs in the presence (■) or absence (□) of lipopolysaccharides (LPS) (10 μg/mL). (**b**) Effect of various concentrations of RS on von Willebrand factor (VWF) expression in HUVECs in the presence (■) or absence (□) of LPS (10 μg/mL). (**c**) Effect of various concentrations of heparin on TF expression in HUVECs in the presence (■) or absence (□) of LPS (10 μg/mL). (**d**) Effect of various concentrations of heparin on VWF expression in HUVECs in the presence (■) or absence (□) of LPS (10 μg/mL). Significant difference in values between the endothelial cells treated with LPS but without SC and the cells treated with LPS and various concentrations of RS is shown as * *p* < 0.01. Significant difference in values between the endothelial cells without SC and the cells treated with various concentrations of RS is shown as **^†^**
*p* < 0.01. The data are cited from [14]. Reproduced with permission from Springer Nature, 2019.

**Figure 10 marinedrugs-18-00228-f010:**
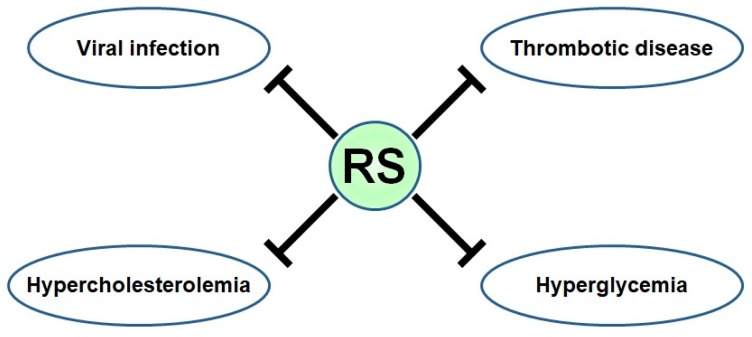
Rhamnan sulfate (RS) extracted from the green algae *Monostroma nitidum* has a wide range of health promotion activities.

**Table 1 marinedrugs-18-00228-t001:** Biological activities of rhamnan sulfate (RS) extract from *Monostroma nitidum*.

Biological Activity of RS	References
Anti-hyaluronidase	in vitro study [23]
Antitumor	in vitro study [22]
Anti-obesity	in vivo study [25]
Anti-thrombosis	in vitro study [14,15,16,17,18,19,20,21], in vivo study [16,17]
Antivirus	in vitro study [6,9,10,11,13], in vivo study [12,13]
Anti-hypercholesterolemia	in vivo study [25], human trial [24]
Anti-hyperglycemia	in vivo study [26], human trial [26]

**Table 2 marinedrugs-18-00228-t002:** Antithrombotic factors expressed in normal vascular endothelium and its functional mechanisms.

Substance	Function		Antithrombotic Mechanism	
TM	Modulator of thrombin activity Anticoagulation and anti-inflammation factor	Suppression of coagulation via protein C activation Enhancement of fibrinolysis via TAFI activation Inhibition of PAMPs/DAMPs and complement factors		
Heparan sulfate	Activator of AT and TFPI	Suppression of coagulation via inhibition of thrombin, FXa, and FVIIa,		
TFPI	Inhibitor of TF, FXa, and FVIIa	Suppression of coagulation initiation via inhibition of TF, FXa and FVIIa		
Protein S	Activator of APC and TFPI	Enhancement of APC-mediated anticoagulation and TFPI-mediated anticoagulation		
PGI_2_	Inhibitor of platelet activation	Inhibition of platelet activation		
ecto-ATP/ADPase	Inhibitor of platelet activation	Inhibition of platelet activation		
NO	Vasodilator	Inhibition of platelet activation and decrease of shear stress		
t-PA	Activator of fibrinolysis	Activation of plasminogen to plasmin		
IL-4	Anti-inflammatory cytokine	Inhibition of vascular endothelial inflammation		
IL-10	Anti-inflammatory cytokine	Inhibition of vascular endothelial inflammation		

TM, thrombomodulin; APC, activated protein C; TAFI, thrombin-activated fibrinolysis inhibitor; PAMPs, pathogen-associated molecular patterns; DAMPs, damage-associated molecular patterns; AT, antithrombin; TFPI, tissue factor pathway inhibitor; NO, nitric oxide; PGI_2_, prostacyclin; t-PA, tissue plasminogen activator; IL-4, interleukin-4; IL-10, interleukin-10.

**Table 3 marinedrugs-18-00228-t003:** Thrombogenetic factors expressed in injured vascular endothelium and its functional mechanisms.

Substance	Function	Thrombus Formation Mechanism
TF	Activator of blood coagulation	Activation of blood coagulation via FVIIa-mediated activation of FX and FIX
Factor V	Cofactor of blood coagulation	Enhancement of FXa-mediated activation of prothrombin
Factor VIII	Cofactor of blood coagulation	Enhancement of FIXa-mediated activation of FX
VWF	Activator of platelet aggregation	Stimulation of platelet aggregation under shear stress
TXA2	Activator of platelet and vasoconstrictor	Activation of platelets and vasoconstriction
PAF	Activator of platelet	Activation of platelets
PAI-1	Plasminogen activator inhibitor	Inhibition of t-PA-mediated plasminogen activation
TNF-α	Inflammatory cytokine	Induction of inflammation, cell adhesion molecules, and apoptosis
ET-1	Vasoconstrictor	Induction of vasoconstriction
Mac-1	Receptor on leukocytes	Binding to intercellular adhesion molecule-1, etc.
E-selectin	Cell adhesion molecule	Binding to leukocytes (neutrophils, monocytes, etc.)

TF, tissue factor; FVII, Factor VII; FIX. Factor IX; FX, Factor X; VWF, von Willebrand factor; TXA2, thromboxane A2; PAF, platelet activating factor; PAI-1, plasminogen activator inhibitor-1; TNF-α, tumor necrosis factor-α; ET-1, endothelin-1; Mac-1, macrophage-1 antigen (integrin α_M_β_2_); E-selectin, endothelial leukocyte adhesion molecule-1.

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
