# Peer review of "Biological Activities of Rhamnan Sulfate Extract from the Green Algae Monostroma nitidum (Hitoegusa)"

_marinedrugs, 2020, doi:10.3390/md18040228_

Round 1

Reviewer 1 Report

The manuscript entitled "Biological activities of rhamnan sulfate extract from the green algae Monostroma nitidum (Hitoegusa)" submitted koji Suzuki et al. describes the review of Rhamnan sulfate and its antithrombotic activity in comparison with heparin. Please see comments below:

1) In section 3.3.4, if blood coagulation activity of RS is less compared to heparin, why RS  is not inhibiting as efficiently as heparin though they are structurally similar? Please explain

2) In fig 8a, what exactly the reason for RS inhibiting thrombin activity less than heparin in the presence of antithrombin? Does the interaction between RS or Heparin and antithrombin leads to any effect? Please give rationale.

3) RS is structurally a sulfated polysaccharide, how does it helps diabetics to suppress the blood glucose level? Rationalize here?

4) Figure 5 illustrates vascular endothelial inflammation leads to thrombotic disorders, how does RS helps minimize these disorders. Explain the role please.

5) It would be great if the authors discusses the antiviral effect of RS as how it works with relevant in vivo study.

Reviewer 2 Report

Revision of manuscript marinedrugs-763545 entitled “Biological activities of rhamnan sulfate extract from the green algae Monostroma nitidum (Hitoegusa)”.

Dear authors,

I believe that the topic of the manuscript is interesting and the manuscript well written. However, I have serious concerns about the current manuscript. Please see below my comments.

Kind regards

  1. A review paper “provide concise and precise updates on the latest progress made in a given area of research”. The current review has 31 references and only 9 of them published in the last 5 years. I will recommend to increase the content of the current manuscript, incorporating tables to summarize key information and to increase considerably the amount of scientific references on this field of study.

  1. There are full sections without any scientific reference and thus, non-acceptable in a review paper. Few of these sections, although this is not an exhaustive list, include: “1. Introduction”, “3.3. Antithrombotic effects of RS”, “3.3.1. Blood coagulation and platelet aggregation under vascular endothelial cell inflammation”.

  1. There are also several non-peer-reviewed papers cited in the manuscript “submitted to Marine Drugs”.

  1. Multiple figures are non-attributed to the original authors. I.e. Figure 6, Figure 7, Figure 8 and Figure 9. The original sources should be cited in the main manuscript and also in the figure caption.

  1. Figure 1. If these images are taken by the researchers please specify the conditions (scale bars and magnification) for images 1.b and 1.c.

  1. In the introduction “Here, we focus on describing the antithrombotic effects of RS, which has been studied extensively”. However, the review focused on multiple properties of these molecules and there are not many references provided. Please delete the other biological properties or modify the focus of the review.

  1. There are many sections that lack of scientific depth. I.e. section “3.1. Antiviral properties” is based on 4 studies, and the same applies to section “3.2. Antiobesity…”

  1. I will recommend expanding the current work and incorporating tables summarizing the biological properties of these compounds and the type of activity and assessment made. I.e. in vitro antiviral activity, ex-vivo antiviral activity, in vivo, human trials…

Author Response

A review paper “provide concise and precise updates on the latest progress made in a given area of research”. The current review has 31 references and only 9 of them published in the last 5 years. I will recommend to increase the content of the current manuscript, incorporating tables to summarize key information and to increase considerably the amount of scientific references on this field of study.

We think the reviewer’s opinions are quite correct. We added Table 1 which shows a summary of biological activities of RS reported to date.

There are full sections without any scientific reference and thus, non-acceptable in a review paper. Few of these sections, although this is not an exhaustive list, include: “1. Introduction”, “3.3. Antithrombotic effects of RS”, “3.3.1. Blood coagulation and platelet aggregation under vascular endothelial cell inflammation”.

The reviewer’s opinions are quite correct. We have added 27 citations as the base of the manuscript. Finally, the paper has 58 references.

There are also several non-peer-reviewed papers cited in the manuscript “submitted to Marine Drugs”.

We deleted non-peer-reviewed papers from the manuscript.

The original data obtained by Terasawa and his group regarding antiviral effects of RS are now submitted to journal of “Marine Drugs” and now on the revised version review stage. Therefore, the descriptions in this manuscript are cited as “submitted to Marine Drugs”.

Multiple figures are non-attributed to the original authors. I.e. Figure 6, Figure 7, Figure 8 and Figure 9. The original sources should be cited in the main manuscript and also in the figure caption.

The data shown in Figure 6, 7, 8, and 9 are cited from our own paper [20]. We cited this paper in the main manuscript and the figure caption.

Figure 1. If these images are taken by the researchers please specify the conditions (scale bars and magnification) for images 1.b and 1.c.

We added scale bars and magnification for images in Figure 1.b and 1.c.

In the introduction “Here, we focus on describing the antithrombotic effects of RS, which has been studied extensively”. However, the review focused on multiple properties of these molecules and there are not many references provided. Please delete the other biological properties or modify the focus of the review.

We think the reviewer’s opinion is reasonable, so we change the manuscript as follows.

Recently, some biological properties of RS have been demonstrated by in vitro and in vivo studies, which probably protect human subjects from viruses and ameliorate vascular dysfunction caused by metabolic disorders, especially lifestyle-related diseases.  In this review, we focus on the antithrombotic effects of RS and introduce its antiviral and other biological activities.

There are many sections that lack of scientific depth. I.e. section “3.1. Antiviral properties” is based on 4 studies, and the same applies to section “3.2. Antiobesity…”

According to your opinion, we added more detail information of biological activities RS on antiviral and anti-obesity as 3.3. section in the manuscript.

I will recommend expanding the current work and incorporating tables summarizing the biological properties of these compounds and the type of activity and assessment made. I.e. in vitro antiviral activity, ex-vivo antiviral activity, in vivo, human trials…

According to the reviewer’s comment, we added Table 1 which shows a summary of biological activities of RS studied in vitro or in vivo to date.

Reviewer 3 Report

This manuscript entitle is that” Biological activities of rhamnan sulfate extract from the green algae Monostroma nitidum (Hitoegusa), the paper is well written, however it needs to be revised in several point.

Major point

  1. Figure 3, the resolution of the picture is too low
  2. Rhamnan sulfate (RS), along with a decription of various activities, would be nice if a schematic or picture was added.
  3. Figure 5, also the resolution of the picture is too low
  4. Figure 4, it would be nice if the picture was clear
  5. A summary of RS for each biological activity is provide, however a full summary is required

Round 2

Reviewer 2 Report

Dear authors, 

The authors did a great job modifying the review, as the current version has been modified according to the comments of the reviewers, more information and literature are currently cited and in my opinion the quality of this version has improved.

My opinion still maintains on citing articles that are not accepted yet, as in the case of rejection, this review will be citing non-peer reviewed literature. I leave this point out of my review as it will have to be evaluated according to the journal and editorial guidelines of Marine drugs and MDPI.

Best regards